# How do the effects of toxicity in competitive online video games vary by source and match outcome?

**Jacob Morrier**[1]*, **Amine Mahmassani**[2], **R. Michael Alvarez**[1]

**1** Division of the Humanities and Social Sciences, California Institute of Technology, Pasadena, California, United States of America, **2** Activision®, Santa Monica, California, United States of America

* jmorrier@caltech.edu

**Data availability statement:** This study analyzes proprietary data collected and owned by Activision, with a detailed description provided in the manuscript. Due to commercial and confidentiality restrictions, this data cannot

## Abstract

This article seeks to estimate variations in the effects of toxicity in competitive online video games by source and match outcome. To this end, we analyze proprietary data from the first-person action video game *Call of Duty®: Modern Warfare®III*, published by Activision®. To overcome causal identification issues, we implement an instrumental variable estimation strategy. Our findings confirm that exposure to toxicity has statistically significant causal effects on short-term player engagement and the probability that players engage in similar behavior in the current match. Further, we show that these effects vary significantly depending on whether toxicity originates from opponents or teammates, whether it originates from teammates in the same or a different party, and the match's outcome. These findings have meaningful implications regarding the allocation of resources for combating toxicity and the nature of toxicity across various contexts.

## Introduction

Competitive online video games are a popular form of entertainment, with approximately 190.6 million players in the United States and 3.4 billion globally [1,2]. While they provide a positive experience to many players, they can also expose them to undesirable behavior, such as bullying, cheating, trolling, and toxicity. According to a 2023 survey, 76% of adult players report having experienced harassment in online multiplayer video games [1]. The incidence of toxic behavior in online multiplayer video games is generally attributable to their competitive nature and the anonymity conferred by online interactions [3–5]. Research indicates that toxicity in competitive online video games has become normalized, with some players perceiving it as an inherent and acceptable aspect of gaming culture, much like in competitive sports [4,6–8]. To put this issue into perspective, video games' massive player bases mean that even a low incidence of toxicity results in thousands of daily incidents, affecting an even higher number of players.

The adverse effects of toxicity are widely acknowledged and well-documented. Two stand out as particularly noteworthy for the video game industry. First, toxicity drives player churn and dissuades new players from joining [7,9–11]. This effect provides a compelling business

be shared publicly. For access inquiries, please contact Gary Quan, Expert Technical Project Manager at Activision®/Demonware, at gquan@demonware.net.

**Funding:** Activision funded this study through a grant, with RMA as the principal investigator. Activision also provided financial support to AM in the form of a salary. Activision collected the data as part of its routine commercial activities but had no involvement in the design of this study, the data analysis, the decision to publish, or the preparation of the manuscript. No other external funding was received for this study.

**Competing interests:** The authors declare no other competing interests.

case for combating toxicity. Indeed, while video game service operators may seek to mitigate toxicity for ethical reasons, such as protecting players from psychological harm and promoting an inclusive and positive gaming environment, the negative effect of toxicity on player engagement highlights their vested interest in combating toxicity since it can ultimately impede the commercial success of their products.

Second, toxicity tends to spread, with exposure to it causing other players to engage in similar behavior [10,12–15]. As the adage goes, humans are, by nature, social beings. Accordingly, their peers heavily influence their actions. A wealth of empirical research, both experimental and observational, has exposed strong correlations and causal relationships between an individual's behavior and outcomes and those of their environment [16–22]. This influence extends to virtuous and objectionable behavior, including academic dishonesty, bullying, and crime. In competitive online video games, the propagation of toxicity amplifies the consequences of a single player's misconduct, increasing the industry's incentives to address the issue before it becomes entrenched.

This article seeks to estimate the magnitude of these effects across different contexts. These estimates carry meaningful implications for the allocation of resources for combating toxicity. With limited available resources, we must direct them where they can have the most impact. In particular, we should target resources to contexts where the undesirable effects of toxicity on player engagement or its proliferation are most pronounced, ensuring that each prevented instance of toxicity brings the highest returns. In contrast, we should redirect resources away from contexts where players find satisfaction in behavior otherwise considered toxic. This issue is especially relevant in competitive online video games, where the boundary between acceptable and unacceptable behavior can often be blurred [8]. In this context, an overall negative effect may conceal positive consequences in some contexts and negative ones in others. By assessing how toxicity affects player engagement in various contexts, we can more effectively distinguish between toxic and acceptable behavior, allowing us to focus resources on combating the former.

We analyze differences in the effects of exposure to toxicity across three dimensions: (i) whether it originates from teammates or opponents, (ii) whether it comes from teammates in the same party, with whom players voluntarily choose to team up, or a different party, and (iii) whether the exposed player's team wins or loses the match. For reference, parties are groups of one or more players who voluntarily choose to play together. The matchmaking algorithm typically keeps these parties together when forming teams. The literature has yet to explore how the effects of exposure to toxicity interact with these factors. These variables are readily observable and, thus, can readily be used to guide the allocation of resources. We expect the nature and effects of toxicity to differ significantly based on these factors.

To achieve our goal, we analyze proprietary data from *Call of Duty*®, a popular first-person action video game franchise published by Activision®. We focus on one of the series' recent installments, *Call of Duty: Modern Warfare*®*III*, particularly its most popular multiplayer mode, Team Deathmatch. In this mode, players are divided into two equally sized teams and compete to achieve the highest number of eliminations. After a brief pause, eliminated players reappear at a different location on the map. A team wins by reaching a predetermined elimination limit first or accumulating the most eliminations by the end of the match.

Since 2023, Activision has partnered with Modulate™, a startup developing intelligent voice technology to identify online toxicity, and incorporated its proprietary voice chat moderation technology, ToxMod™, into its gaming platforms [23]. ToxMod is a voice moderation technology that analyzes in-game voice chat interactions based on features such as transcribed content, volume, emotion, and intention [24]. These features are fed into machine learning models to detect six types of toxic content: adult language, audio assaults, cultural hate speech,

sexual hate speech, sexual vulgarity, and violent speech. This technology's beta rollout began in North America on August 30, 2023, within *Call of Duty: Modern Warfare II* and *Call of Duty: Warzone™*, followed by a global release (excluding the Asia-Pacific region) coinciding with the launch of *Call of Duty: Modern Warfare III* on November 10, 2023. ToxMod only supported English during our observation period.

ToxMod provides unique data on toxicity and players' exposure to it, serving as the basis of our analysis. Our dataset consists of data from a subset of matches in Team Deathmatch mode monitored by ToxMod during the first month after the game's release. We classify a player as having engaged in toxicity if ToxMod flagged at least one of their voice chat interactions as toxic during a match.

We perform two regression analyses. The first considers the effect of exposure to toxicity from opponents and teammates depending on whether the exposed player's team wins or loses the match. The second considers the effect of exposure to toxicity from teammates in a different party and the same party—teammates assigned algorithmically or those with whom players voluntarily teamed up, respectively—depending on whether the exposed player's team wins or loses the match. In both analyses, we estimate the effect of exposure to toxicity on two outcome variables: (i) the time players take to enter their next match as a measure of short-term player engagement, and (ii) the likelihood that exposed players use toxic language in the current match as a measure of the contemporaneous propagation of toxicity.

Even with a large volume of high-quality data, analysts seeking to estimate the causal effect of exposure to toxicity face considerable statistical challenges. The reason is that, in observational data, some variables not accounted for in our regression models—because they are unmeasured or unmeasurable, for instance—may be simultaneously correlated with players' outcomes and their exposure to toxicity, a phenomenon known as endogeneity [25,p. 513]. For example, teammates may concomitantly use toxic language in reaction to a random event occurring in a match, which might also influence their short-term player engagement. More fundamentally, players mutually affect each other. As a result, whether a player, their teammates, and their opponents engage in toxicity is jointly determined. Ultimately, endogeneity introduces biases in standard ordinary least squares (OLS) estimates and obscures the cause-to-effect relationship of exposure to toxicity. No previous observational study on toxicity in competitive online video games has addressed this causal identification issue.

One way to address endogeneity is with randomized controlled experiments. However, due to ethical and logistical constraints, conducting an experiment that randomly exposes players to toxicity is impossible. Instead, we propose an identification strategy neutralizing the causal identification issues in the available observational data. We implement an instrumental variable or two-stage least squares (2SLS) estimation strategy that leverages the fact that we observe players participating in multiple matches with different players. With this strategy, we isolate variations in outcomes of interest caused by interactions with players who, in prior matches with other players, have employed toxic language more frequently and, consequently, are more likely to use such language in the current game. This approach allows us to reliably assess whether and, if so, to what extent exposure to toxicity *causes* variations in player engagement and their likelihood of using similar language, distinguishing our findings from the existing literature.

## Hypotheses

We formulate five hypotheses regarding the effects of exposure to toxicity depending on its source and the match outcome:

H1. Toxicity from teammates has a weaker effect on player engagement than toxicity from opponents.

H2. Toxicity from teammates spreads more than toxicity from opponents.

H3. Toxicity from teammates in the same party has a weaker effect on player engagement than toxicity from teammates in a different party.

H4. Toxicity from teammates in the same party spreads more than toxicity from teammates in a different party.

H5. Toxicity has a weaker effect when the exposed player's team wins the match.

There are strong theoretical justifications for these hypotheses. In general, players are less likely to engage in harmful behavior toward teammates, as they share common goals and interests, unlike opponents, whose interests directly conflict with their own. Evidence that cooperation between players reduces aggression in video games supports this assertion [26–28]. In this context, we expect that players are less likely to direct toxicity at teammates, particularly those in the same party. Even when players expose teammates to toxicity as bystanders rather than victims, it is more likely to be perceived as innocent and, consequently, should have a weaker effect on player engagement, if any [8]. Conformity to social norms is one of the primary explanations for peer effects [29]. In general, individual perceptions of these norms are influenced more strongly by those with whom they feel a stronger affinity and connection [30,31]. This should apply to teammates, particularly those in the same party, increasing the likelihood that players will mirror their behavior. The same principle holds if social learning is supposed to drive the spread of toxicity. Finally, a player's team winning may reduce the effects of toxicity, as success can foster emotional regulation and strengthen psychological resilience [32].

Previous studies generally support these hypotheses. First, they provide evidence that players are more prone to hostile behavior when their teammates, particularly their friends, engage in such actions, suggesting that contagion is more pronounced in these contexts [12, 28,33]. However, other studies find that exposure to toxicity from opponents is associated with a larger increase in the likelihood that they engage in similar behavior, highlighting a retaliatory response [14]. Studies indicate that some players prefer retreating when confronted with toxic players [7]. Also, while playing with friends can increase engagement, long-term player retention is negatively affected by playing with toxic friends, particularly for veteran players [9]. Finally, many studies show that toxic behavior is more prevalent when a team is losing, suggesting that players may resort to toxic behavior under these circumstances they do not otherwise [12,34–36].

## Data and methodology

### Dataset description

Our dataset contains data from a subset of matches in Team Deathmatch mode monitored by ToxMod from November 10 to December 10, 2023. Our sample is not comprehensive, as it only includes matches monitored through ToxMod in a single game mode, among other limitations. It consists of 56,464,489 observations, each representing a player in a game, from 4,167,325 matches and 4,539,599 players. On average, we observe each player participating in 12.44 games.

This data reflects gameplay during the first month after the game's launch and may not reflect activity later in its lifecycle. Early on, an influx of new players may need time to familiarize themselves with the game, including the toxicity prevailing in gaming culture. Accordingly, it might take some time before new players engage in toxicity [14]. Even veteran players may need time to adapt to newly introduced features. Finally, seasonal factors can significantly affect gameplay, with activity typically peaking during holidays and tapering off as daylight hours increase [37].

Due to technical issues, exposure data is missing for 34.6% of speech acts flagged as toxic by ToxMod. Fig 1 displays the daily evolution of the share of unavailable exposure data throughout our observation window. From November 17, one week after the game's launch, exposure data became inaccessible for some offenses. Thereafter, the daily share of missing exposure data fluctuated between 5 and 73%, with 35 to 65% of exposure data unavailable on most days.

If exposure data is missing randomly and, in particular, independently of the source of toxicity and the match outcome, it does not introduce bias in our findings. It might still dilute our coefficients' magnitude, but the small probability of exposure to toxicity suggests that this dilution is negligible. While demonstrating that exposure data is randomly missing is difficult, it seems plausible given the disruption's cause. To support this conjecture, we present two pieces of evidence that the proportions between exposure probabilities conditional on variables of interest remain unchanged despite the missing exposure data. It follows that exposure data is missing in roughly similar proportions regardless of whether toxicity came from an opponent or a teammate, whether the teammate was in the same party or a different one, or whether the exposed player's team won or lost.

First, Figs 2 and 3 illustrate the daily evolution of the probability that players are exposed to toxicity in different contexts throughout our study's timeframe. A dashed vertical line

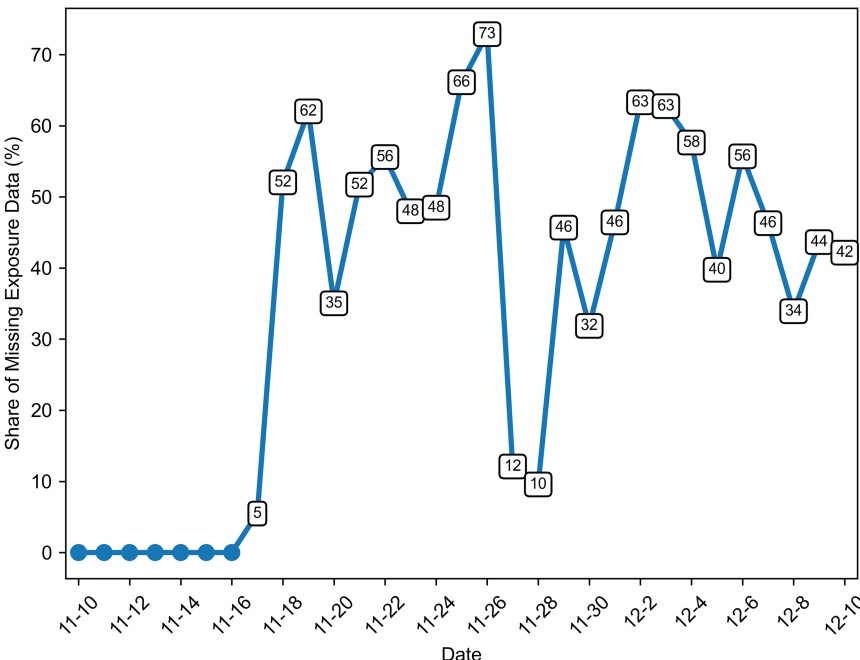

**Fig 1. Daily evolution of the share of missing exposure data.**

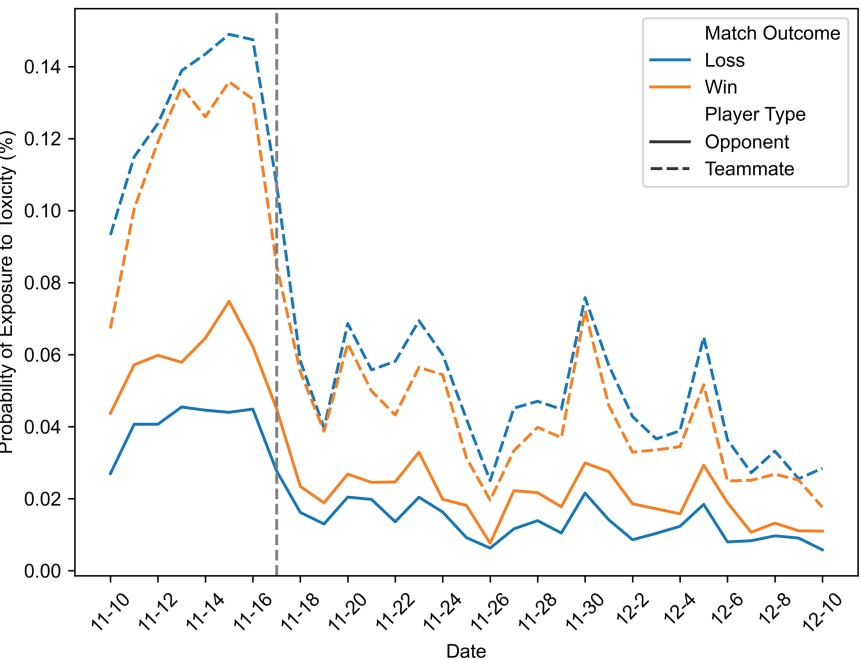

**Fig 2. Daily evolution of the probability of exposure to toxicity from opponents and teammates.**

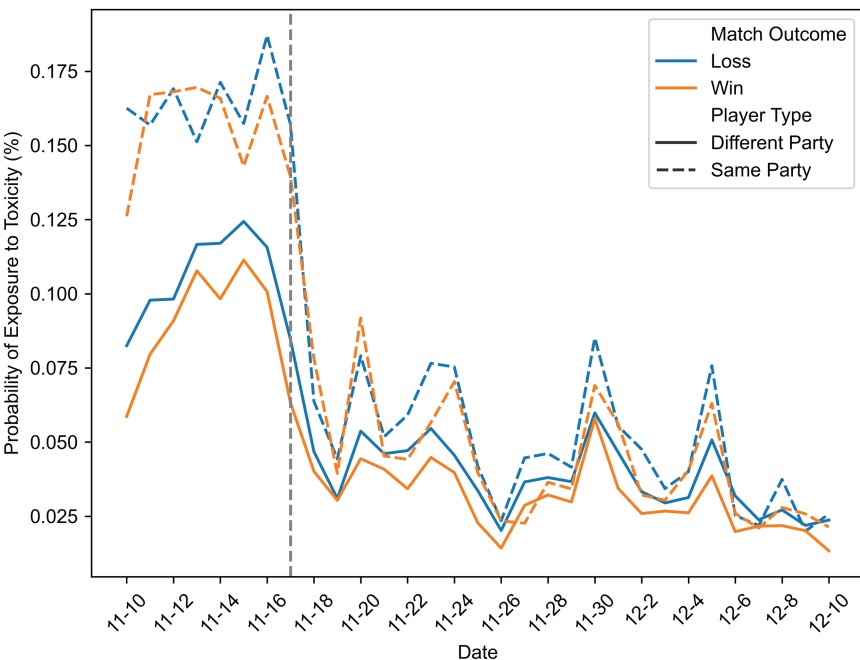

**Fig 3. Daily evolution of the probability of exposure to toxicity from teammates in a different party and the same party.**

marks the day after which some exposure data becomes unavailable. The proportions between exposure probabilities in various contexts are roughly constant throughout our observation window, including before and after November 17. Consequently, the missing data does not significantly alter the observed exposure patterns.

Second, Figs 4 and 5 illustrate the probability of a player being exposed to toxicity from opponents or teammates, whether in the same party or a different party, depending on whether the player's team won or lost during the period from March 4 to April 12, 2024. Over this period, we have comprehensive exposure data for a random subset of matches. This figure indicates that the proportions between exposure probabilities in different contexts, as illustrated in Figs 6 and 7, are consistent in our observation window and a later period during which we have exhaustive exposure data.

## Model specification

In this article, we seek to estimate the causal effect exposure to toxicity has on player engagement and their probability of using such language. We define these effects as the variation in the average time players take to enter their next match and the likelihood that they use

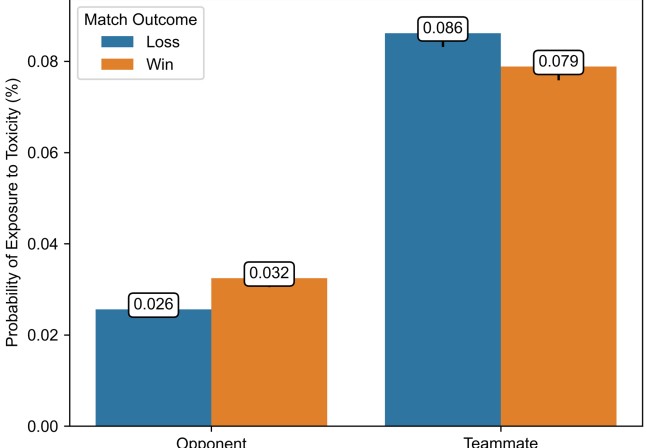

**Fig 4. Probability of exposure to toxicity from opponents and teammates from March 4 to April 12, 2024.**

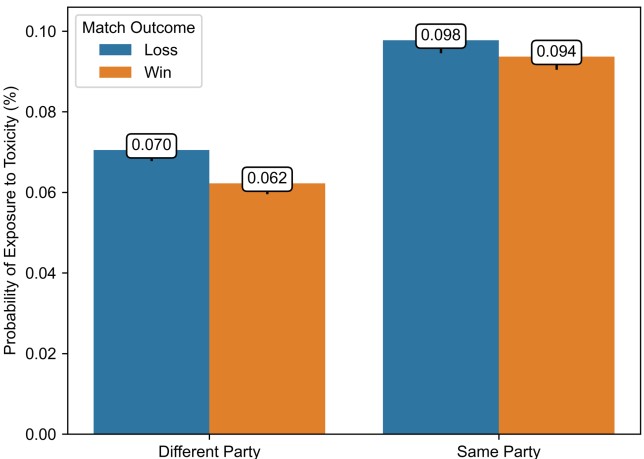

**Fig 5. Probability of exposure to toxicity from teammates in a different party and the same party from March 4 to April 12, 2024.**

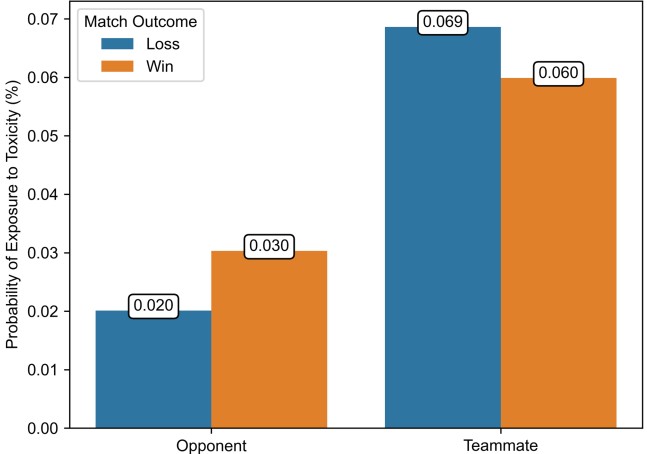

**Fig 6. Probability of exposure to toxicity from opponents and teammates.**

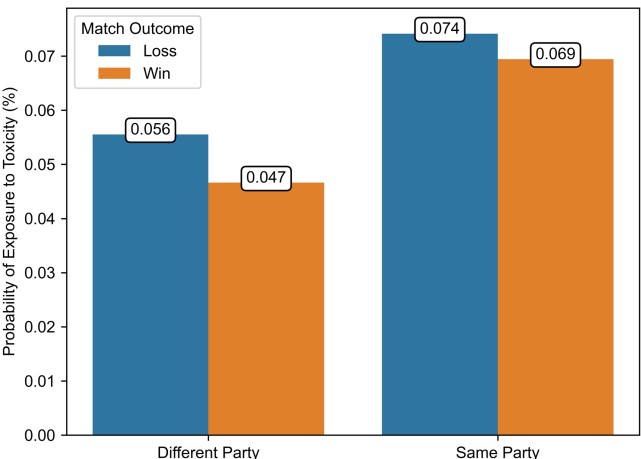

**Fig 7. Probability of exposure to toxicity from teammates in a different party and the same party.**

toxic language in the current game, respectively, caused by exposure to toxicity from another player, holding all other variables constant.

In light of this, we define the following structural model of players' behavior:

$$y_{ij} = \alpha_j + \beta \cdot \boldsymbol{x}_{ij} + \varepsilon_{ij},$$

where:

- $y_{ij}$ is the outcome of interest in match $i$ for player $j$.
- $\alpha_j$ is a player-specific intercept.
- $\beta$ is a coefficient vector.
- $\boldsymbol{x}_{ij}$ is a covariates vector.
- $\varepsilon_{ij}$ is an error term.

In this model, the outcomes of interest are the time players take to enter their next match and their probability of using toxic language in the current game. The covariates include the

number of teammates and opponents—or teammates from the same party and a different party, depending on the model specification—who expose player $j$ to toxicity in match $i$, the outcome of match $i$ for player $j$'s team, and interactions between these variables. For reference, Table 1 lists the covariates included in each model specification.

Our structural model posits that outcomes of interest are primarily affected by two factors: (i) their intrinsic tendency to exhibit the outcome of interest, and (ii) the number of other players who expose them to toxicity. The coefficients $\beta$ reflect the causal effect of exposure to toxicity on outcome variables. They are the estimands of our analysis.

Before addressing causal identification issues, let us first clarify what the time players take to enter their next match captures. Consider a player who ends their current session and plans to return at the same hour the next day. In this scenario, 24 hours will elapse before their next match. Conversely, if a player joins a new match immediately, the elapsed time will be nearly zero. Overall, the time players take to enter their next match captures the interaction of two factors: (i) the probability they end their current session after a match, and (ii) the interval before they return to start a new session.

## Causal identification issues

Naturally, one might consider estimating the coefficients $\beta$ using OLS. However, contrary to the standard assumptions in linear regression models, the covariates are not independent of the error terms, resulting in endogeneity.

Endogeneity stems from various sources, each posing a threat to the causal identification of our estimands. One source is model misspecification, as some variables are omitted from the model because they are either unmeasured or unmeasurable. These omitted variables may simultaneously affect the outcomes of interest and the likelihood of being exposed to toxicity. For example, endogeneity might occur if a player and their teammates resort to toxicity in response to an exogenous event in the game, with this random event also affecting the time they take to enter their next match.

Self-selection poses another threat to causal identification. Players sometimes form parties to engage in toxicity or under the expectation that their party members will do so. Also, when two players decide to join forces, it suggests a degree of familiarity between them. This familiarity can change the dynamics of their interactions, influencing both their likelihood of using toxic language and their chances of being exposed to toxicity through one another. In parallel, it may affect their level of engagement, causing them to enter their next match more quickly. When players do not voluntarily team up, their previous interactions can still have a lasting impact.

Endogeneity mechanically arises when estimating the effect of exposure to toxicity on the probability that a player uses toxic language. The reason is that players in a match mutually

**Table 1. Model covariates.**

| Model specification I | Model specificiation II |
| --- | --- |
| Number of teammates who expose the player to toxicity | Number of teammates in the same party who expose the player to toxicity |
| Number of opponents who expose the player to toxicity | Number of teammates in a different party who expose the player to toxicity |
| | Binary variable indicating whether the player is in a party with other players |
| Binary variable indicating whether the player's team won or lost the match | |
| *Interactions between variables* | |

influence each other. To illustrate, consider a simplified scenario where a player has only one teammate and no opponents. In this case, the dependent variable in some equations appears on the right-hand side of others. Thus, the use of toxic language by players and their teammates is interdependent and jointly determined.

Formally, let us consider the pair formed by players $j$ and $k$ in match $i$. The two equations determining whether these players engage in toxicity are:

$$Y_{ij} = \alpha_j + \beta Y_{ik} + \varepsilon_{ij}$$
$$Y_{ik} = \alpha_k + \beta Y_{ij} + \varepsilon_{ik}.$$

To show that $Y_{ik}$ and $\varepsilon_{ij}$ are correlated, we substitute the first equation into the second and rearrange the resulting expression to isolate $Y_{ik}$ on the left-hand side:

$$Y_{ik} = \alpha_k + \beta \left( \alpha_j + \beta Y_{ik} + \varepsilon_{ij} \right) + \varepsilon_{ik} \Leftrightarrow \left( 1 - \beta^2 \right) Y_{ik} = \alpha_k + \beta \left( \alpha_j + \varepsilon_{ij} \right) + \varepsilon_{ik}$$
$$\Leftrightarrow Y_{ik} = \frac{\beta}{1 - \beta^2} \left( \alpha_j + \varepsilon_{ij} \right) + \frac{1}{1 - \beta^2} \left( \alpha_k + \varepsilon_{ik} \right).$$

This equation implies that the error term $\varepsilon_{ij}$ directly enters the value of $Y_{ik}$, resulting in a correlation between them. Intuitively, this means that OLS estimates capture a teammate's effect on a player's inclination to engage in toxicity and its "reflection," that is, the influence this player exerts on their teammate.

## Identification strategy

To address the issues outlined above, we define an identification strategy leveraging the fact that we observe players participating in multiple matches with different players. Our approach is to implement an instrumental variable or 2SLS estimation strategy, a standard causal identification strategy. In particular, we instrument the variables representing the number of teammates and opponents exposing the player to toxicity in the current match with the sum of their probabilities to have used toxic language *in prior matches with other players*. This strategy isolates variations in outcome variables caused by interactions with players who, in previous matches with other players, have had a greater tendency to engage in toxicity and, therefore, are more likely to use such language in the current game.

Henceforth, for tractability, we consider a model that treats exposure to toxicity uniformly, regardless of its source. This model has a single coefficient reflecting the average effect of one other player engaging in toxicity. We can readily extend our approach to differentiate between sources of toxicity.

Formally, our identification strategy consists of adding the following equation to our structural model of players' behavior:

$$x_{ij} = \delta_j + \gamma \sum_{k \in \mathcal{P}_{i,-j}} \sum_{\ell \in \mathcal{M}_{i,k,-j}} \frac{x_{\ell k}^\star}{\#\mathcal{M}_{i,k,-j}} + u_{ij},$$

where:

- $\mathcal{P}_{i,-j}$ is the set of players in match $i$ excluding player $j$.
- $\mathcal{M}_{i,k,-j}$ is the set of matches prior to match $i$ to which player $k$ but not player $j$ participated.
- $x_{\ell k}^\star$ is a binary variable indicating whether player $k$ used toxic language in match $\ell$.
- $u_{ij}$ is an error term.

The instrumental variable is computed by summing over all players other than player $j$ in match $i$, indexed by $k$, the probability with which they have used toxic language in previous matches they participated in without player $j$, indexed by $\ell$. This instrumental variable belongs to the general class of spatial or "leave-one-out" instruments introduced in empirical industrial organization for demand and supply estimation and commonly used for the causal identification of simultaneous equation models [38,39].

For an instrument to be valid, it must satisfy two conditions: (i) relevance, meaning that there must be a strong correlation between the instrumental and endogenous explanatory variables, and (ii) exclusion, meaning that the instrumental variables must be independent of the structural model's error term. We can empirically verify the validity of the first condition by examining the estimates of the first-stage regressions. As a rule of thumb, the $F$ statistic against the null hypothesis that the instruments are irrelevant in the first-stage regressions should have a value greater than ten. Table 2 presents the coefficients for the instrumental and exogenous explanatory variables and the $F$ statistic for all first-stage regressions in our analysis. Each column corresponds to an endogenous explanatory variable, and each row represents an instrumental or exogenous explanatory variable. For all first-stage regressions, the $F$ statistic significantly exceeds ten, indicating a strong first stage.

On the other hand, we cannot empirically test the validity of the exclusion restriction. Instead, it depends on the assumptions we are ready to make regarding the relationship between the instrumental variables and the structural equation's error term. We argue that

**Table 2. First-stage regression estimates.**

**(A) Opponents and Teammates.**

|  | Opponents | Teammates | Opponents × Win | Teammates × Win |
|---|---|---|---|---|
| Opponents | 0.0082*** | 0.0015*** | −0.0003*** | −0.0005*** |
|  | (0.000) | (0.000) | (0.000) | (0.000) |
| Teammates | −0.0001 | 0.0330*** | −0.0003*** | −0.0012*** |
|  | (0.000) | (0.001) | (0.000) | (0.000) |
| Opponents × Win | 0.0047*** | −0.0012*** | 0.0156*** | 0.0014*** |
|  | (0.001) | (0.000) | (0.001) | (0.000) |
| Teammates × Win | 0.0007** | −0.0012 | 0.0009*** | 0.0345*** |
|  | (0.000) | (0.001) | (0.000) | (0.001) |
| Win | 0.0001*** | 0.0000*** | 0.0002*** | 0.0005*** |
|  | (0.000) | (0.000) | (0.000) | (0.000) |
| $F$ Statistic | 365.2 | 801.8 | 1615.0 | 3159.9 |

**(B) Teammates in a different party and the same party.**

|  | Different Party | Same Party | Different Party × Win | Same Party × Win |
|---|---|---|---|---|
| Different Party | 0.0271*** | −0.0001* | −0.0003*** | −0.0001*** |
|  | (0.001) | (0.000) | (0.000) | (0.000) |
| Same Party | −0.0010** | 0.0359*** | −0.0004** | −0.0045*** |
|  | (0.000) | (0.003) | (0.000) | (0.001) |
| Different Party × Win | −0.0002 | −0.0001 | 0.0275*** | −0.0000 |
|  | (0.001) | (0.000) | (0.001) | (0.000) |
| Same Party × Win | −0.0002 | −0.0005 | −0.0002 | 0.0449*** |
|  | (0.001) | (0.004) | (0.000) | (0.003) |
| Win | −0.0001*** | −0.0000*** | 0.0004*** | 0.0001*** |
|  | (0.000) | (0.000) | (0.000) | (0.000) |
| Player is in a Party | −0.0000* | 0.0007*** | −0.0000 | 0.0003*** |
|  | (0.000) | (0.000) | (0.000) | (0.000) |
| $F$ Statistic | 505.1 | 917.4 | 2051.4 | 605.2 |

*Note:* $^*p < 0.1$; $^{**}p < 0.05$; $^{***}p < 0.01$

calculating the instrument with the probability of a player using toxic language in previous matches with other players neutralizes the principal sources of endogeneity.

First, the fact that no data from the current match enters the instrumental variables neutralizes endogeneity caused by events occurring in the current game that simultaneously affect the outcomes of interest and exposure to toxicity. For instance, it addresses the case wherein a player and one or more of their teammates use toxic language in reaction to, say, one of their common opponents using such language or another exogenous event.

Second, the fact that no data from the other matches wherein both players participated enters the instrumental variables neutralizes endogeneity from enduring factors reflecting their relationship and simultaneously affecting outcomes of interest and their exposure to toxicity, including but not exclusively through each other.

Third, using only data from past matches to compute the instrumental variables neutralizes the long-term effects of exposure to toxicity on the outcomes of interest. This is especially important when estimating the effect of exposure to toxicity on a player's probability of using such language. Indeed, whether player $j$ uses toxic language in a match may affect the propensity of one of the other players, say, player $k$, to use such language in future matches, regardless of whether player $j$ participates in it. More generally, all events in the current game may influence players' future behavior. Consequently, if data from future matches entered the instrumental variables, it would open a "backdoor" for a player's use of toxic language or other events in the current game to penetrate the instrument, thereby violating the exclusion restriction.

In interpreting our findings, we must keep in mind that our estimation strategy provides an estimate of the local average treatment effect for "compliers," defined as those players who were exposed to toxicity because they interacted with other players more likely to use toxic language in previous matches with other players and, consequently, exogenously more likely to use such language in the current game. Compliers do not include players who seek to alter their exposure to toxic language by intentionally deactivating the voice chat to evade it or using toxic language to provoke reactions from other players, for instance. If the effect of exposure to toxicity is heterogeneous, this local average treatment effect might not accurately reflect the average treatment effect for the entire player population.

## Estimation

Our model contains player-specific intercepts, also called fixed effects, capturing the inherent tendency of players to exhibit outcomes of interest. Estimation of these fixed effects is computationally expensive. Consequently, analysts frequently resort to "down-sampling," which consists of sampling a computationally convenient number of observations and estimating the model with fixed effects only for those. This results in a lower statistical accuracy.

Another method exists to overcome the computational cost of estimating fixed effects. Explicitly estimating the fixed effects is superfluous since they are not directly relevant to our analysis. Our reason for including them in the model is to absorb time-invariant variables affecting individual players' propensity to exhibit the outcomes of interest. This is critical if there is a correlation between a player's inherent tendency to display the outcomes of interest and their likelihood of being exposed to toxicity.

We can achieve the same end by demeaning the values of the dependent, independent, and instrumental variables for all players at the individual level [40,p. 427]. Upon doing so, we estimate the coefficients $\beta$ through the standard 2SLS estimation procedure without resorting to any down-sampling.

We restrict our analysis to observations for which: (i) we observe at least one other player in the current match play at least one other match with other players so that we can compute the value of the instruments for them, and (ii) we observe the player participate in at least two matches so that we can demean the values of the dependent, independent, and instrumental variables for them. These restrictions result in some attrition.

## Ethical considerations

Caltech's Institutional Review Board reviewed and granted an exemption for this study (Approval number: IR23-1395). It does not involve participants prospectively recruited by the authors. The data was collected as part of Activision's routine commercial activities and does not include any information that could identify individual participants.

## Results

Regression estimates are presented in Table 3. The effects of exposure to toxicity are illustrated in Figs 6, 7, 8, 9, 10, and 11. A summary of these effects, along with the average values of the outcome variables, is presented in Table 4. A discussion of these findings follows.

### Probability of exposure to toxicity

As a preamble, we consider the probability of exposure to toxicity. Figs 6 and 7 illustrate the likelihood of exposure to toxicity conditional on the match outcome, defined as whether

**Table 3. Regression estimates.**

| | Time to enter the next match | Probability of using toxic language |
|---|---|---|
| **(A) Opponents and teammates.** | | |
| Opponents | 60.683*** | 0.1382*** |
| | (9.4202) | (0.0244) |
| Teammates | 16.182*** | 0.0854*** |
| | (2.8458) | (0.0099) |
| Opponents × Win | −36.596*** | −0.0832*** |
| | (10.781) | (0.0281) |
| Teammates × Win | 0.8545 | −0.0074 |
| | (3.8833) | (0.0152) |
| Win | −0.3794*** | −0.0001*** |
| | (0.0053) | (0.0000) |
| *F* Statistic | 8,958.2 | 504.1 |
| **(B) Teammates in a different party and the same party.** | | |
| Different Party | 17.936*** | 0.0208*** |
| | (3.0740) | (0.0078) |
| Same Party | 12.255* | 0.6949*** |
| | (7.1299) | (0.0785) |
| Different Party × Win | 5.0996 | −0.0045 |
| | (4.3649) | (0.0111) |
| Same Party × Win | −18.705** | −0.0956 |
| | (7.5382) | (0.0910) |
| Win | −0.3811*** | −0.0001*** |
| | (0.0048) | (0.0000) |
| Player is in a Party | −0.4193*** | 0.0013*** |
| | (0.0082) | (0.0000) |
| *F* Statistic | 12,551.2 | 8,454.4 |
| *Note:* *$p < 0.1$; **$p < 0.05$; ***$p < 0.01$ | | |

the exposed player's team won or lost. Fig 6 distinguishes between exposure to toxicity from opponents and teammates, and Fig 7 between toxicity from teammates in a different party and those in the same party.

The probability of exposure to toxicity from opponents or teammates is less than one-tenth of a percent. While some exposure data is missing, implying that the exposure to toxicity could be higher, this suggests that ToxMod primarily identifies the most severe instances of toxicity.

Players are considerably more likely—two to over three times more likely, depending on the match's outcome—to be exposed to toxicity from teammates than opponents. Furthermore, players are more likely to be exposed to toxicity from opponents when their team wins and teammates when their team loses. Among teammates, players are slightly more likely to be exposed to toxicity from those in the same party. However, this difference in the probability of exposure to toxicity from teammates in a different party and the same party is much smaller than the difference in the likelihood of exposure to toxicity from opponents and teammates.

## Effects of toxicity from opponents and teammates

We now turn to the effect of exposure to toxicity on the time players take to enter their next match and their probability of using similar language in the current game. Figs 8 and 9 illustrate the estimated effect of exposure to toxicity from opponents and teammates conditional on the match's outcome. Figs 8 and 9 depict the effect of exposure to toxicity on the time players take to enter their next match and the probability that players use toxic language in the current game, respectively. Each estimate represents the average marginal effect of exposure to toxicity from one player. We illustrate estimates with their 95% confidence interval.

Exposure to toxicity significantly increases the time before players enter their next match. This effect ranges from 16.18 to 60.68 hours, depending on whether the toxicity originates from opponents or teammates and whether the exposed player's team won or lost. This delay is substantial compared to the average time players take to enter their next match, which is 3.66 hours when their team wins and 4.17 hours when their team loses. Therefore, exposure to

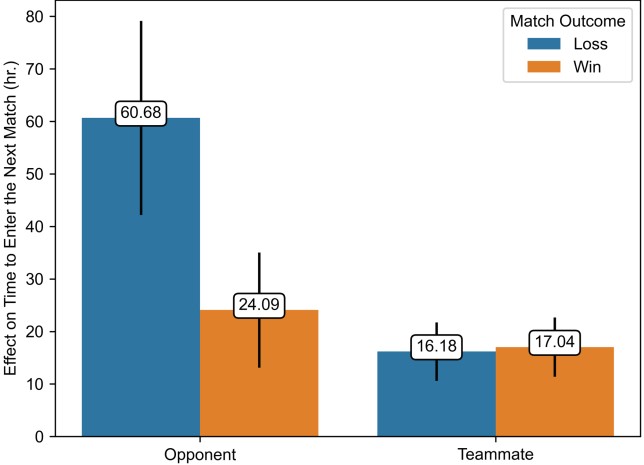

**Fig 8. Effects of exposure to toxicity from opponents and teammates on the time players take to enter their next match.**

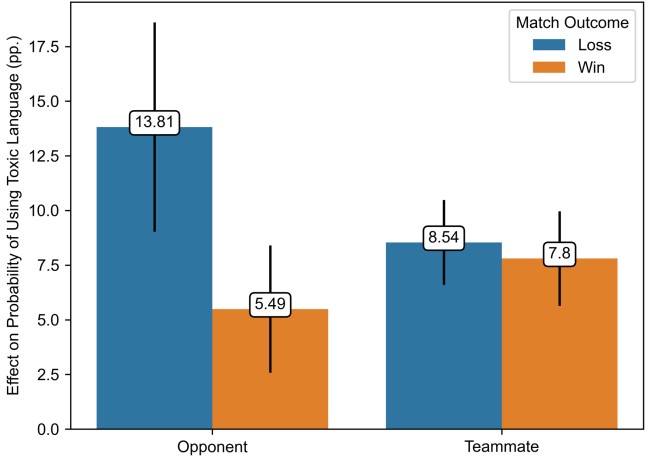

**Fig 9. Effects of exposure to toxicity from opponents and teammates on the probability that players use toxic language in the current match.**

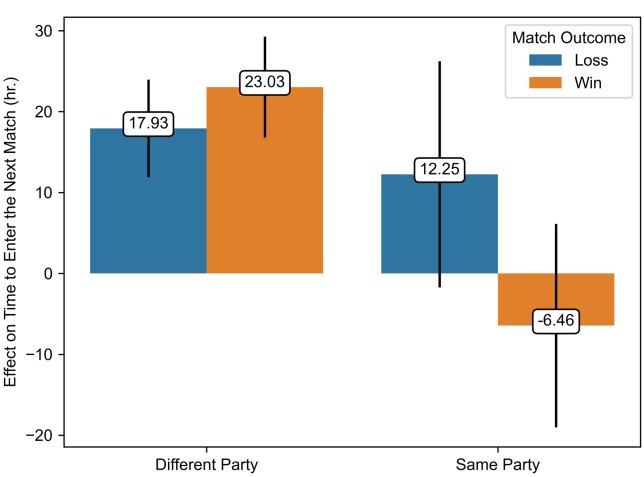

**Fig 10. Effects of exposure to toxicity from teammates in a different party and the same party on the time players take to enter their next match.**

toxicity increases by a factor of five to 16 the time players take to enter their next match. The effect of exposure to toxicity from opponents is significantly higher when the exposed player's team loses, corroborating Hypothesis 5. Conversely, exposure to toxicity from teammates has a higher effect when the exposed player's team wins, though this difference is not statistically significant. The most pronounced effect is caused by exposure to toxicity from opponents when the player's team loses. The effect of exposure to toxicity from opponents when the player's team wins is much smaller. The latter is slightly higher in magnitude—although not significantly different—than the effect of exposure to toxicity from teammates regardless of the match's outcome. On the whole, these findings support Hypothesis 1. Exposure to toxicity also significantly increases the probability that a player uses similar language. This effect ranges from 5.49 to 13.81 percentage points, depending on whether toxicity originates from opponents or teammates and whether the player's team won or lost. This effect is considerable, given that the observed incidence of toxicity is 0.078% when the exposed player's team

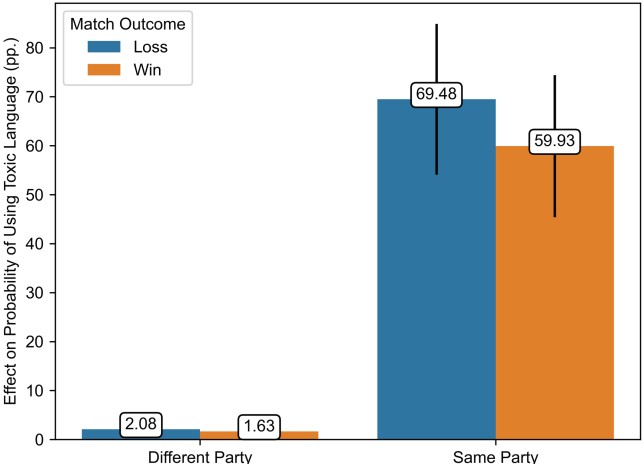

**Fig 11. Effects of exposure to toxicity from teammates in a different party and the same party on the probability that players use toxic language in the current match.**

**Table 4. Effects of exposure to toxicity.**

| Source of toxicity | Outcome variable | | | |
| --- | --- | --- | --- | --- |
| | Time to enter the next match (hrs.) | | Probability of using toxic language (pp.) | |
| | Match outcome | | | |
| | Win | Loss | Win | Loss |
| Opponents | 24.09 | 60.68 | 5.49 | 13.81 |
| Teammates | 17.04 | 16.18 | 7.8 | 8.54 |
| Teammates in a different party | 23.03 | 17.93 | 2.08 | 1.63 |
| Teammates in the same party | –6.46 (ns.) | 12.25 (ns.) | 69.48 | 59.93 |
| *Average* | *3.66* | *4.17* | *0.078* | *0.088* |

*Note:* ns. denotes estimates that are not statistically significant at the 95% confidence level.

wins and 0.088% when it loses. Irrespective of its source, the effect of exposure to toxicity is higher when the player's team loses, corroborating Hypothesis 5. The most pronounced effect is caused by exposure to toxicity from opponents when the player's team loses. Conversely, the least pronounced effect is caused by exposure to toxicity from opponents when the player's team wins. The latter is significantly smaller than the former. Exposure to toxicity from teammates exerts an effect of intermediate value on the probability that a player uses similar language. Accordingly, Hypothesis 2 is partially verified, at least in the context of the propagation of toxicity when the exposed player's team loses.

## Effects of toxicity from different-party and same-party teammates

Figs 10 and 11 illustrate the estimated effect of exposure to toxicity from teammates in a different party and those in the same party conditional on the match's outcome. Figs 10 and 11 depict the effect of exposure to toxicity on the time players take to enter their next match and the probability that a player uses toxic language in the current game, respectively. Each estimate represents the average marginal effect of exposure to toxicity from one player.

Exposure to toxicity from teammates in a different party significantly increases the time before players enter their next match. This effect is substantial, with a delay of 17.94 hours

after a loss and 23.04 hours after a win, equivalent to multiplying by a factor of five to seven times the time players take to enter their next match. In contrast, regardless of the match's outcome, exposure to toxicity from teammates in the same party does not significantly affect the time players take to enter the next game, supporting Hypothesis 3. In particular, when the exposed player's team wins, the effect of toxicity from teammates in the same party is negative and, thereby, significantly smaller than the effect of exposure to toxicity from teammates in a different party. These findings partially support Hypothesis 5, at least regarding the impact of exposure to toxicity from teammates in the same party.

Exposure to toxicity also significantly increases the probability that players adopt similar language. The effect of exposure to toxicity from teammates in the same party is particularly pronounced, resulting in a 59.93 to 69.48 percentage point increase in the probability that a player uses toxic language depending on the match's outcome. In contrast, exposure to toxicity from teammates in a different party has a much smaller effect, with a magnitude of 1.63 to 2.08 percentage points depending on the match's outcome. These results corroborate Hypothesis 4. Furthermore, all else equal, toxicity spreads more when the exposed player's team loses than when it wins, supporting Hypothesis 5.

## Discussion and conclusion

Our analysis provides valuable insights into the effect of exposure to toxicity on the time before players enter their match and their likelihood of using similar language. Our findings confirm that toxicity significantly affects player engagement, often negatively. Moreover, toxicity spreads as players exposed to it become more likely to use similar language. These results highlight the video game industry's vested interest in combating toxicity.

We show that the effects of exposure to toxicity vary significantly with its source—whether it originates from opponents, teammates from a different party, or teammates in the same party—and the match's outcome. The findings broadly validate our hypotheses. They also have practical implications, guiding video game service operators in targeting their efforts to combat toxicity. Specifically, to minimize the adverse effects of toxicity on player engagement, our analysis advises allocating resources for combating toxicity in the following order of decreasing priority:

1. Toxicity from opponents when the player's team loses.
2. Toxicity from opponents when the player's team wins.
3. Toxicity from teammates in a different party when the player's team wins.
4. Toxicity from teammates in a different party when the player's team loses.
5. Toxicity from teammates in the same party when the player's team loses.
6. Toxicity from teammates in the same party when the player's team wins.

On the other hand, to minimize the proliferation of toxic language, our analysis advises allocating resources for combating toxicity in the following order of decreasing priority:

1. Toxicity from teammates in the same party when the player's team loses.
2. Toxicity from teammates in the same party when the player's team wins.
3. Toxicity from opponents when the player's team loses.
4. Toxicity from opponents when the player's team wins.
5. Toxicity from teammates in a different party when the player's team loses.
6. Toxicity from teammates in a different party when the player's team wins.

These recommendations diverge depending on whether the primary objective is to minimize the negative effect of exposure to toxicity on player engagement or the propagation of toxicity. If the priority is to mitigate the impact of toxicity on player engagement, addressing toxicity from opponents should be a priority. In contrast, toxicity from teammates in the same party has a minimal effect on player engagement. Based solely on this factor, it may not deserve any intervention since its effect is not statistically significant. On the other hand, if our priority is to limit the proliferation of toxicity, addressing toxicity from teammates in the same party becomes a priority since it contributes most to its propagation.

Our findings also have meaningful implications regarding the nature of toxicity in different contexts. We find that exposure to toxicity has a lower effect on player engagement when it comes from teammates, particularly those in the same party as the exposed player when their team wins the match. In parallel, players are more likely to join the bandwagon. These findings suggest that toxicity from teammates, particularly those in the same party, is less likely to be directed at players when their team wins. There is only one other scenario in which exposure to toxicity has a higher effect on the likelihood that players engage in similar behavior, namely when the toxicity comes from opponents and the exposed player's team loses the match. In this case, players likely retaliate against their opponents' toxicity. Remarkably, this behavior is less common when the exposed player's team wins, possibly because the victory provides a sense of retribution on its own.

Admittedly, this study presents some limitations. As noted above, some exposure data is missing. Although we are confident it does not introduce biases in our findings, we cannot demonstrate it with certainty. In addition, our analysis focuses on a single game and game mode. Our results may not extend to other games and modes, particularly those beyond first-person action video games, let alone entirely different settings such as social networks and online forums. We also focus on one form of toxicity: toxic language in voice chat interactions. It excludes other expressions of toxicity, including toxic language in text chat interactions, that may occur in competitive online video games.

In conclusion, our work paves the way for exciting research. First, while our analysis focuses on the short-term effects of exposure to toxicity, there is limited evidence of its long-term impact. Addressing this gap would require data spanning a longer timeframe. We should also consider how the effects of exposure to toxicity differ based on factors beyond its source and the match outcome, including players' experience, skill levels, and cultural influences. Examining a broader range of games and game modes across various genres would help overcome the limitations discussed earlier. Finally, although we have identified where the video game industry should target its resources and interventions, additional evidence is needed regarding the effectiveness of various strategies for preventing toxic behavior to determine the industry's optimal course of action in these situations [41].

## Acknowledgments

The authors thank Andrea Boonyarungsrit, Grant Cahill, Min Kim, Rafal Kocielnik, Jonathan Lane, Zhuofang Li, Gary Quan, Deshawn Sambrano, Feri Soltani, Carly Taylor, and Michael Vance for their invaluable feedback and support in writing this article.

## Author contributions

**Conceptualization:** Jacob Morrier, Amine Mahmassani.

**Data curation:** Jacob Morrier.

**Formal analysis:** Jacob Morrier.

**Funding acquisition:** R. Michael Alvarez.

**Investigation:** Jacob Morrier, Amine Mahmassani.

**Methodology:** Jacob Morrier, Amine Mahmassani.

**Project administration:** R. Michael Alvarez.

**Software:** Jacob Morrier.

**Supervision:** Amine Mahmassani, R. Michael Alvarez.

**Validation:** Jacob Morrier, Amine Mahmassani.

**Visualization:** Jacob Morrier.

**Writing – original draft:** Jacob Morrier.

**Writing – review & editing:** Jacob Morrier, Amine Mahmassani, R. Michael Alvarez.

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
