## [Decision Letter · Decision Letter 0]

26 Feb 2025

PONE-D-24-33185Uncovering the Effect of Toxicity on Player Engagement and its Propagation in Competitive Online Video GamesPLOS ONE

Dear Dr. Morrier,

Thank you for submitting your manuscript to PLOS ONE. After careful consideration, we feel that it has merit but does not fully meet PLOS ONE’s publication criteria as it currently stands. Therefore, we invite you to submit a revised version of the manuscript that addresses the points raised during the review process.

We look forward to receiving your revised manuscript.

Kind regards,

Bernard Fong

Academic Editor

PLOS ONE

**Journal Requirements:**

1. When submitting your revision, we need you to address these additional requirements. Please ensure that your manuscript meets PLOS ONE's style requirements, including those for file naming. The PLOS ONE style templates can be found at https://journals.plos.org/plosone/s/file?id=wjVg/PLOSOne_formatting_sample_main_body.pdf and https://journals.plos.org/plosone/s/file?id=ba62/PLOSOne_formatting_sample_title_authors_affiliations.pdf 2. Please update your submission to use the PLOS LaTeX template. The template and more information on our requirements for LaTeX submissions can be found at http://journals.plos.org/plosone/s/latex. 3. Thank you for stating in your Funding Statement: Activision provided funding for this study through a sponsored research grant. Please provide an amended statement that declares *all* the funding or sources of support (whether external or internal to your organization) received during this study, as detailed online in our guide for authors at http://journals.plos.org/plosone/s/submit-now.  Please also include the statement “There was no additional external funding received for this study.” in your updated Funding Statement. Please include your amended Funding Statement within your cover letter. We will change the online submission form on your behalf. 4. Thank you for stating the following in the Competing Interests section:  AM contributed to this article while being employed by Activision. The opinions expressed by the authors do not represent the views of Activision. The other authors state that they have no competing interests.We note that one or more of the authors are employed by a commercial company: Activision.  a. Please provide an amended Funding Statement declaring this commercial affiliation, as well as a statement regarding the Role of Funders in your study. If the funding organization did not play a role in the study design, data collection and analysis, decision to publish, or preparation of the manuscript and only provided financial support in the form of authors' salaries and/or research materials, please review your statements relating to the author contributions, and ensure you have specifically and accurately indicated the role(s) that these authors had in your study. You can update author roles in the Author Contributions section of the online submission form. Please also include the following statement within your amended Funding Statement. “The funder provided support in the form of salaries for authors [insert relevant initials], but did not have any additional role in the study design, data collection and analysis, decision to publish, or preparation of the manuscript. The specific roles of these authors are articulated in the ‘author contributions’ section.”If your commercial affiliation did play a role in your study, please state and explain this role within your updated Funding Statement.  b. Please also provide an updated Competing Interests Statement declaring this commercial affiliation along with any other relevant declarations relating to employment, consultancy, patents, products in development, or marketed products, etc.   Within your Competing Interests Statement, please confirm that this commercial affiliation does not alter your adherence to all PLOS ONE policies on sharing data and materials by including the following statement: "This does not alter our adherence to  PLOS ONE policies on sharing data and materials.” (as detailed online in our guide for authors http://journals.plos.org/plosone/s/competing-interests) . If this adherence statement is not accurate and  there are restrictions on sharing of data and/or materials, please state these. Please note that we cannot proceed with consideration of your article until this information has been declared. Please include both an updated Funding Statement and Competing Interests Statement in your cover letter. We will change the online submission form on your behalf. 5. In the online submission form you indicate that your data is not available for proprietary reasons and have provided a contact point for accessing this data. Please note that your current contact point is a co-author on this manuscript. According to our Data Policy, the contact point must not be an author on the manuscript and must be an institutional contact, ideally not an individual. Please revise your data statement to a non-author institutional point of contact, such as a data access or ethics committee, and send this to us via return email. Please also include contact information for the third party organization, and please include the full citation of where the data can be found. 6. Your ethics statement should only appear in the Methods section of your manuscript. If your ethics statement is written in any section besides the Methods, please move it to the Methods section and delete it from any other section. Please ensure that your ethics statement is included in your manuscript, as the ethics statement entered into the online submission form will not be published alongside your manuscript.

**Additional Editor Comments:**

Both reviewers believe the paper has good potentials and have pointed out some areas of improvements. The authors are suggested to systematically address the reviewers' comments and resubmit their revised manuscript.

Reviewers' comments:

Reviewer's Responses to Questions

**Comments to the Author**

1. Is the manuscript technically sound, and do the data support the conclusions?

Reviewer #1: Partly

Reviewer #2: Yes

2. Has the statistical analysis been performed appropriately and rigorously? 

Reviewer #1: Yes

Reviewer #2: Yes

3. Have the authors made all data underlying the findings in their manuscript fully available?

Reviewer #1: No

Reviewer #2: No

4. Is the manuscript presented in an intelligible fashion and written in standard English?

Reviewer #1: No

Reviewer #2: Yes

5. Review Comments to the Author

**Reviewer #1:** General comment:

This manuscript provides a well-structured investigation into the impact of toxicity on player engagement within online gaming, using "Call of Duty: Modern Warfare III" as a case study. The present study intends to investigate the influence of toxic language on players’ level of engagement and their use of similar toxic language under the setting of online video gaming. While the study put specific focus on the toxicity in a first-person action video gaming shows some originality that is commendable in its design and methodological rigor, there are critical areas that require further clarification, elaboration, and adjustments to enhance the robustness and clarity of the findings. Specifically, there are several major issues that are necessary to be addressed, including the confusing logic flow of the review of literature, lack of representative studies in the literature review part, unclear conceptual framework, lack of hypothesis establishment, and lack of proper academic English writing. Besides, no line numbers are provided in the manuscript makes it more difficult to read and review. Therefore, the aforementioned fundamental issues should be further revised. I hope my comments below help to clarify what I am thinking and shed insight on some ways to tackle these problems moving forward.

Introduction:

Lack of background information: The primary issue of this manuscript is the lack of necessary background information of the context, which is the use of toxic languages and behaviors in online gaming settings, in justifying the rationale of conducting this study. The authors elaborated three reasons to reinforce the significance of this study, while all these three reasons lack empirical evidence to support the justifications. For example, in the beginning of this piece of work, the authors could first demonstrate the current development of online gaming (especially first-person action video games) by using empirical statistics to reflect its development in recent years as well as the significant influence on players. Then, the authors might refer to some evidence from existing articles to further articulate the universal phenomenon of toxicity in online gaming which could also lead to negative effects and behaviors. After that, the authors can move to the further elaboration of the estimates’ effects on consequent variables, including the exposure of toxicity, player engagement, and probability to use a similar toxic language. The authors should discuss each hypothetical relationship one by one and provide empirical justifications to support the hypotheses.

Literature Gaps: Another major issue in the introduction part is the lack of thorough review of current literature relating to the topic. While the introduction provides a rationale, it would benefit from an explicit identification of specific gaps in the literature that this study seeks to address. Detailing gaps can provide clearer research positioning. When discussing the effect of exposure to toxic language in the online gaming on player engagement and the consequent use of similar toxic language, for example, the authors should make a fruitful review of existing literature to support the hypothesis, indicating that the establishment of such hypothesis is theoretically supported. For example, when discussing the exposure to toxic language on player engagement, the authors should define what is toxic language and player engagement. After that, the authors should elaborate on previous literature focusing on use of toxic language’s influence on engagement in other areas, such as team sports and other gaming settings. Then, based on these discussions, the authors can finally arrive at the hypothetical relationship between the exposure of toxic language and the online gaming engagement. Also, please establish a clear statement to indicate your hypothetical relationship based on previous discussion, such as “H1-1: The exposure of toxic language in the first-person online gaming can significantly influence player’s gaming engagement.” These are the essential elements which should be included in this quantitative study.

Finally, the third and fourth paragraph in the page 3 and the second paragraph in page 4 should be placed in the method section, which illustrates the research techniques that were implemented in this study. Moreover, it’s better to clear statement the research purpose and the brief significance of the study at the end of introduction section. In addition, a conceptual framework to show the overall hypothetical causal relationship among variables is necessary to be depicted.

Method:

Dataset Scope and Representation: While it’s noted that the dataset contains data from "Call of Duty: Modern Warfare III" over one month, the representativeness of this timeframe needs elaboration. For example, is this month indicative of typical player engagement, or could seasonal factors (like game launches or holidays) influence toxicity levels? Including a rationale for selecting this timeframe would support the study's generalizability.

Moreover, the dataset description mentions that exposure data is unavailable for 34.6% of toxic statements. It would be valuable to expand on why this data is unavailable and how the absence might affect outcome reliability. For instance, is there any indication that missing data is systematically related to certain types of matches or player demographics? Clarifying this could strengthen the data’s integrity.

Rationale for Instrumental Variables (IV): The choice of instrumental variables, especially the “leave-one-out” approach, is intriguing but requires further justification. The authors could provide a brief comparison with alternative approaches, explaining why IV is particularly suited for isolating toxicity's causal effect on player engagement.

Explanation of Variables: The notation used to specify variables, such as yij for outcome, αj for intercept, and vector xij, might benefit from clearer definitions. A table listing each variable and its description, along with how each represents the model’s components, would improve accessibility for readers less familiar with econometric modeling.

Down-Sampling Implications: Since the dataset contains millions of observations, a down-sampling technique was employed. Expanding on how down-sampling was conducted—whether through a random sample, stratified by certain variables, or other means—would clarify whether the down-sampled data accurately represents the overall dataset. Furthermore, discussing whether down-sampling could introduce bias in estimated effects is essential. For instance, is there a possibility that high-intensity matches or those with certain toxicity patterns are underrepresented? Addressing this could include a sensitivity analysis to check the robustness of results with varying down-sample sizes.

Ethical issue: It is also necessary to provide a detailed explanation of the ethical review process and the decision, including the ethical application reference number, the procedures, and documents submitted for ethical review.

Results:

Summary of findings: The authors should include one separate paragraph in the beginning of the results section of stating the summary of the findings after the analysis. This guiding paragraph can signal the reader with a general understanding of the following elaboration.

Magnitude of Effect on Match Re-Entry Times: The reported delay in re-entering the next match (up to 60.68 hours in some contexts) requires further interpretation. This substantial effect size suggests strong aversion, but additional analysis explaining the practical implications of this delay is needed. For instance, how does this delay impact overall player retention or leave? Offering more justifications or comparing this delay to normal re-engagement times in other contexts could contextualize its significance.

Summarizing Key Findings in a Table: A summary table that consolidates the main findings—especially the effect sizes for different sources of toxicity and contexts (opponent, same-party, different-party)—would be a helpful quick reference. This table could include columns for source of toxicity, effect on re-engagement time, effect on propagation probability, and statistical significance.

Discussion:

Integration with Existing Literature: While the discussion synthesizes the study’s findings, it would benefit from a deeper integration with prior research on toxic behavior in online environments and social contagion theory. Specifically, linking findings on toxic behavior propagation to similar behaviors observed in social networks, online forums, or even real-world sports settings would situate this study within a broader context. This could also include comparisons to other studies that have observed peer influence in different online gaming contexts.

Explaining Mechanisms of Toxicity Propagation: The study reveals that exposure to toxicity, particularly within same-party contexts, has a strong effect on propagation. Expanding on the psychological or social mechanisms that may drive this phenomenon—such as social learning, peer influence, or in-group favoritism—could provide theoretical insight. Including references to theories of group behavior or behavioral modeling in competitive settings would add depth to the discussion.

Theoretical contributions and practical implications: Based on the results and analysis, the authors should include one paragraph in elaborating the theoretical contributions to the current research area, such as the research gap it has filled, the problems it has answered, or the knowledge it can provide for the existing literature. Besides, another paragraph illustrating the practical implications for practitioners is also a must. The discussion could go further by offering specific, actionable recommendations. For instance, the authors could suggest targeted moderation strategies, such as real-time intervention when toxicity is detected in same-party contexts or offering player rewards for positive behavior, particularly after a loss when players are more vulnerable to engaging in toxicity. Moreover, the study’s prioritization framework for addressing toxicity (e.g., focusing on same-party versus different-party toxicity) is insightful. However, it would benefit from further elaboration on how resources should be allocated. Discussing how game developers could prioritize resources for monitoring and intervention based on match outcomes, party composition, or other specific contexts could provide more practical guidance.

Limitations and Future Research Directions: While the study has a solid methodology, there are inherent limitations that should be explicitly acknowledged. For instance, the reliance on one game and a specific subset of player interactions (e.g., Team Deathmatch mode in “Call of Duty: Modern Warfare III”) may limit generalizability to other gaming environments or demographics. Acknowledging these limitations can provide readers with a balanced understanding of the study’s scope.

Data Availability Constraints: The missing exposure data should be revisited in the limitations. While some technical reasons for this missing data are noted, discussing the potential bias this introduces—particularly if certain types of matches or players are more affected by missing data—would reinforce transparency. The authors could suggest that future studies should ensure a more complete data capture or explore alternative methods to account for missing data.

Future Research Opportunities: This study opens several avenues for future research, and the discussion should explicitly highlight them. Examples might include:

• Cross-Game Comparisons: Studies that compare toxicity propagation across various game types or genres could validate whether findings are unique to the competitive first-person shooter genre or generalizable to other online gaming contexts.

• Longitudinal Effects: Future research could investigate the long-term effects of repeated exposure to toxicity. For instance, does continuous exposure reduce player engagement over months or lead to permanent changes in behavior?

• Intervention Efficacy: Follow-up studies could test the effectiveness of different intervention techniques, such as muting toxic players or rewarding positive behavior. Examining these interventions’ impact on player engagement and community health could provide actionable insights for developers.

By addressing these areas, particularly with greater emphasis on actionable recommendations for game developers and a nuanced consideration of social and psychological mechanisms, the manuscript will offer a more robust and practically relevant contribution to the field. Given the current gaps, a major revision is recommended to ensure that the manuscript meets its full potential in terms of academic rigor, practical applicability, and theoretical significance.

**Reviewer #2: **In this manuscript, the authors present the effect of toxic language on player engagement based on a popular online game (Call of Duty: Modern Warfare III). To proceed with the data analysis, the authors a large pool of data (>50million observations, >4million matches and >4million players) and used ToxMod for monitoring.

Overall, the authors have done a great work to correlate, among others, the toxic language with the average time of a player entering a new match which is crucial for preserving engagement of large amounts of players.

In online gaming and generally in virtual environments we can identify various types of fun (Lazzaro, 4 keys to fun). In online games, like COD, we can find hard fun (“serial” and competivive winners) and people fun (https://www.nicolelazzaro.com/the4-keys-to-fun/). “Hard fun”-ners are the most competitive players, while people fun”-ners are the players who join to play with their friends and enjoy more the social interaction within the game experience. A first correlation that would expand this study more is the level/skill/winning ratio (=> hardcore gamers) along with their tendency to toxic language. Maybe the hard core gamers (“hard fun”-ners) although they face this toxicity by other players (both teammates and opponents) this is is a part of the competitiveness element. An example of real-life sports was the great Michael Jordan who was a famous trash-talker. Trash-talking was a tool that allowed him to both assert his dominance and amplify his competitive edge. (https://www.basketballnetwork.net/off-the-court/why-michael-jordan-only-respected-trash-talk-when-the-score-was-tied). This pushed him further to be a better player and gave him extra motivation. Same principles apply here too.

That being said, although toxicity is considered generally a bad thing it may offer some opposite results in engagement for certain types of players. This is something not addressed in this paper and it would be important if we are trying to be more holistic.

On the other hand, “people fun” type of players may feel discouraged with all this toxicity and less engaged over time. I suspect they would also have a lower level/skillset than the previous type of players. It is worth checking this parameter too.

So, it would be great if the authors had this data and could correlate this info. Their models of understanding the effects of toxicity would be much more deeper, since the total outcome would be the product of the (obviously) negative effects of toxicity but with also its positive engagement results in specific areas/player types.

One more suggestion is to examine the cultural approach of this issue and its effects. Due to technical reasons, like ping/latency, players are usually located in servers somehow close to them. I do not know if it would be possible based on the data already acquired, but it would really interesting both for scientific and industry reasons if the toxic speech affects the same way players’ engagement from geographic locations (e.g North/Western Europe vs Mediterranean players, or US areas).

All in all, the authors have made a great work analyzing this data and creating the models. But this problem is a more complex one and it is a deeper multi-variable problem. One-size-fits-all solutions may raise some parts of engagement but may drop others so, it would be important to be mentioned and if possible researched too.

6. PLOS authors have the option to publish the peer review history of their article (what does this mean?). If published, this will include your full peer review and any attached files.

Reviewer #1: **Yes: **SHI YUCHEN

Reviewer #2: No

---

## [Author Response · Author response to Decision Letter 1]

21 Mar 2025

Attached is our detailed response to the reviewers' comments.

---

## [Decision Letter · Decision Letter 1]

14 May 2025

How do the effects of toxicity in competitive online video games vary by source and match outcome?

PONE-D-24-33185R1

Dear Dr. Morrier,

We’re pleased to inform you that your manuscript has been judged scientifically suitable for publication and will be formally accepted for publication once it meets all outstanding technical requirements.

Kind regards,

Bernard Fong

Academic Editor

PLOS ONE

Additional Editor Comments (optional):

Both reviewers are satisfied that the authors have addressed their concerns in the previous round and the paper is now in a form suitable for publication.

Reviewers' comments:

Reviewer's Responses to Questions

**Comments to the Author**

1. If the authors have adequately addressed your comments raised in a previous round of review and you feel that this manuscript is now acceptable for publication, you may indicate that here to bypass the “Comments to the Author” section, enter your conflict of interest statement in the “Confidential to Editor” section, and submit your "Accept" recommendation.

Reviewer #1: All comments have been addressed

Reviewer #2: All comments have been addressed

2. Is the manuscript technically sound, and do the data support the conclusions?

Reviewer #1: Yes

Reviewer #2: Yes

3. Has the statistical analysis been performed appropriately and rigorously? 

Reviewer #1: Yes

Reviewer #2: Yes

4. Have the authors made all data underlying the findings in their manuscript fully available?

Reviewer #1: Yes

Reviewer #2: No

5. Is the manuscript presented in an intelligible fashion and written in standard English?

Reviewer #1: Yes

Reviewer #2: Yes

6. Review Comments to the Author

Reviewer #1: (No Response)

Reviewer #2: The authors have proceeded with the necessary changes and I suggest that their manuscript be published

7. PLOS authors have the option to publish the peer review history of their article (what does this mean?). If published, this will include your full peer review and any attached files.

Reviewer #1: No

Reviewer #2: No

---

## [Editor Report · Acceptance letter]

PONE-D-24-33185R1

PLOS ONE

Dear Dr. Morrier,

I'm pleased to inform you that your manuscript has been deemed suitable for publication in PLOS ONE. Congratulations! Your manuscript is now being handed over to our production team.

Kind regards,

on behalf of

Dr. Bernard Fong

Academic Editor

PLOS ONE